# Structural basis for virulence regulation in *Vibrio cholerae* by unsaturated fatty acid components of bile

Justin T. Cruite[1,2], Gabriela Kovacikova[3], Kenzie A. Clark [4,5], Anne K. Woodbrey[2,4], Karen Skorupski[3] & F. Jon Kull[1,2,4]*

The AraC/XylS-family transcriptional regulator ToxT is the master virulence activator of *Vibrio cholerae*, the gram-negative bacterial pathogen that causes the diarrheal disease cholera. Unsaturated fatty acids (UFAs) found in bile inhibit the activity of ToxT. Crystal structures of inhibited ToxT bound to UFA or synthetic inhibitors have been reported, but no structure of ToxT in an active conformation had been determined. Here we present the 2.5 Å structure of ToxT without an inhibitor. The structure suggests release of UFA or inhibitor leads to an increase in flexibility, allowing ToxT to adopt an active conformation that is able to dimerize and bind DNA. Small-angle X-ray scattering was used to validate a structural model of an open ToxT dimer bound to the cholera toxin promoter. The results presented here provide a detailed structural mechanism for virulence gene regulation in *V. cholerae* by the UFA components of bile and other synthetic ToxT inhibitors.

---

[1] Department of Biochemistry and Cell Biology, Geisel School of Medicine at Dartmouth College, Hanover, NH, USA. [2] Guarini School of Graduate and Advanced Studies, Dartmouth College, Hanover, NH, USA. [3] Department of Microbiology and Immunology, Geisel School of Medicine at Dartmouth College, Hanover, NH, USA. [4] Department of Chemistry, Dartmouth College, Hanover, NH, USA. [5] Present address: Department of Chemistry, Princeton University, Princeton, NJ, USA. *email: f.jon.kull@dartmouth.edu

Toxigenic *Vibrio cholerae* causes disease by producing the primary virulence factors cholera toxin (CT) and the toxin coregulated pilus (TCP). The transcription of CT (*ctx*) and TCP (*tcpA*) is activated by the AraC/XylS-family transcriptional regulator ToxT[1–8]. AraC/XylS proteins regulate a variety of cellular processes in bacteria, including carbon metabolism, stress response, and virulence[9]. Members of this family of transcriptional activators are defined by a DNA-binding domain containing two helix-turn-helix DNA-binding motifs. Most AraC/XylS-family members, including ToxT and AraC, contain an N-terminal regulatory/dimerization domain and a C-terminal DNA-binding domain. While full-length ToxT has not been shown to form a dimer, the regulatory domain, when separated from the rest of the protein, has been shown to dimerize in vivo, suggesting that the interaction between the two domains somehow regulates dimerization[10]. Dimerization of ToxT is necessary to bind DNA and activate the expression of virulence genes and pairs of ToxT-binding sites have been found in the promoters of virtually all genes activated by ToxT[10–15]. Furthermore, virstatin, an inhibitor of ToxT activity, inhibits the dimerization of ToxT and mutations that render ToxT resistant to virstatin also increase its dimerization[16,17].

Unsaturated fatty acids (UFAs) present in bile reduce the expression of virulence genes in *V. cholerae* without affecting the expression of ToxT[18,19]. The crystal structure of full-length ToxT from epidemic *V. cholerae* serotype O1 El Tor (ToxT$_{EPI}$) was determined previously[20] and was the first full-length structure to be reported for an AraC/XylS-family member with a domain arrangement similar to AraC. Unlike the crystal structure of the AraC dimerization domain, full-length ToxT$_{EPI}$ was monomeric. Unexpectedly, the UFA *cis*-palmitoleate co-purified and crystallized with ToxT$_{EPI}$, bound within the hydrophobic pocket at the interface between the regulatory and DNA-binding domains, at a site analogous to the arabinose-binding pocket of AraC. The carboxylate head of the UFA was bound by two lysine residues, one from the regulatory domain (Lys31) and one from the DNA-binding domain (Lys230), and a tyrosine (Tyr12). Interestingly, the bent conformation of the UFA within the binding pocket of ToxT closely resembled the shape of virstatin. Subsequent experiments showed that UFAs inhibit ToxT$_{EPI}$ DNA-binding in vitro and reduce ToxT-dependent virulence gene expression[20–22]. An alanine substitution of Lys230 was reported previously to increase ToxT-dependent expression of *ctx* nearly twofold, suggesting that the lysine is involved in negatively regulating ToxT activity[23]. Surprisingly, the same mutation had no effect on the expression of *tcpA* in response to UFA or inhibitors[24]. However, the carboxylate moiety of synthetic compounds designed to mimic the shape of a UFA bound to ToxT was shown to be necessary to inhibit ToxT-dependent *tcpA* expression, emphasizing the importance of the UFA-binding lysines in inhibiting ToxT activity[24].

A number of non-O1/non-O139 isolates of *V. cholerae* from the environment that cause outbreaks of gastroenteritis in humans have been found to possess variants of ToxT (ToxT$_{ENV}$) that have a divergent N-terminal domain and are resistant to bile and virstatin[10,25–27]. The DNA-binding domains of these variants share 98–99% sequence identity with ToxT$_{EPI}$. However, the regulatory domains of the variants are only 64–67% identical to ToxT$_{EPI}$. Interestingly, when purified, one of the ToxT variants, ToxT$_{ENV256}$ from environmental *V. cholerae* isolate SCE-256, was observed to have increased solubility compared to ToxT$_{EPI}$.

Since the initial structure of UFA-bound ToxT$_{EPI}$ was determined, several additional structures of ToxT$_{EPI}$ in an inhibitor bound state have been reported[28,29]. However, no structure of ToxT in an active state had been determined. In this study, utilization of the more soluble ToxT$_{ENV256}$ allowed the purification and crystallization of mutants not possible with ToxT$_{EPI}$. We

present here the crystal structures of the *V. cholerae* master virulence activator ToxT in both the UFA-bound and apo states. The structures reveal conformational changes that occur upon the activation of ToxT. In addition, small-angle X-ray scattering has been used to validate a structural model of the ToxT dimer bound to the *ctx* promoter and provide insight into the structure of a fully active ToxT dimer bound to DNA.

## Results

**Crystal structure of ToxT$_{ENV256}$.** We solved the 1.8 Å resolution crystal structure of wild-type ToxT from *V. cholerae* serogroup O42 strain SCE-256 (ToxT$_{ENV256}$) (Fig. 1a). The asymmetric unit is composed of a monomer of ToxT$_{ENV256}$ in a closed conformation. ToxT$_{ENV256}$ and ToxT$_{EPI}$ are superposable, with a root-mean-square deviation (RMSD) of 0.432 Å for 204 α-

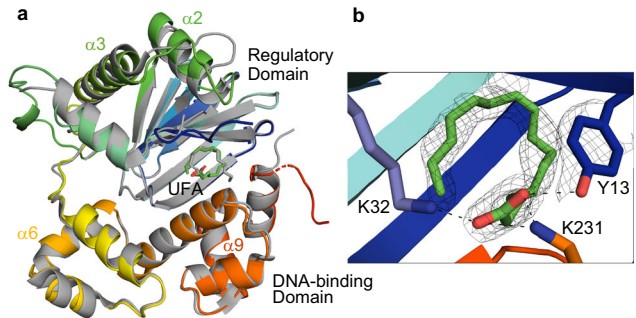

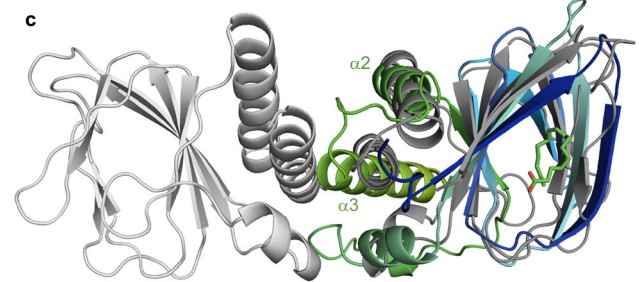

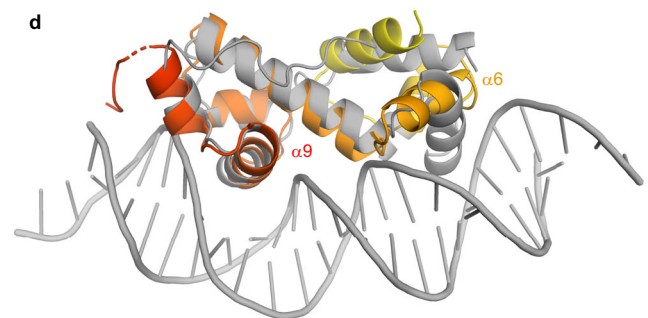

**Fig. 1** Structure of ToxT$_{ENV256}$–unsaturated fatty acid (UFA) complex. **a** Asymmetric unit of the ToxT$_{ENV256}$ (PDB 6P7R) structure aligned with the structure of ToxT$_{EPI}$ (3GBG). ToxT$_{ENV256}$ is colored from the N-terminus to the C-terminus in dark blue to red. ToxT$_{EPI}$ is colored gray. **b** Close-up of the UFA-binding pocket of ToxT$_{ENV256}$ showing the sidechain interactions with the carboxylate head. Electron density is shown as the 2Fo-Fc map contoured to 1.5 σ. **c** Structural alignment of the regulatory domain of UFA-bound ToxT$_{ENV256}$ to the AraC regulatory domain dimer (PDB 2ARA). AraC is colored gray, the regulatory domain of UFA-bound ToxT$_{ENV256}$ is colored blue to green. **d** Structural alignment of the DNA-binding domain of UFA-bound ToxT$_{ENV256}$ to MarA in complex with DNA (PDB 1BL0). MarA is colored gray, the DNA-binding domain of UFA-bound ToxT$_{ENV256}$ is colored yellow to red.

carbons. The N-terminal regulatory domain contains a nine sheet β-barrel with three α helices on one face. The C-terminal domain is entirely α-helical and contains two helix-turn-helix DNA-binding motifs. As with ToxT$_{EPI}$, ToxT$_{ENV256}$ purified from *E. coli* with a UFA bound within the hydrophobic pocket inside the end of the regulatory domain β-barrel, at the interface between the regulatory domain and the DNA-binding domain (Fig. 1b). Tyr13, Lys32, and Lys231 in ToxT$_{ENV256}$ are analogous to Tyr12, Lys31, and Lys230 in ToxT$_{EPI}$. As seen in the structure of ToxT$_{EPI}$, the carboxylate head of the UFA forms interactions with the sidechains of Tyr13, Lys32, and Lys231 of ToxT$_{ENV256}$.

Structural alignment of the regulatory domains of UFA-bound ToxT$_{ENV256}$ and AraC (Fig. 1c), and the DNA-binding domain of UFA-bound ToxT$_{ENV256}$ with the AraC-family member MarA in complex with DNA (Fig. 1d), suggests a mechanism for the allosteric inhibition of ToxT dimerization and DNA-binding by UFA[30,31]. Helix α3 of ToxT is analogous to the helix that forms the homodimer interface of AraC. When aligned to the structure of the AraC dimer, helix α3 of UFA-bound ToxT is at an angle that precludes dimerization. In the MarA–DNA complex structure, the recognition helices of the two helix-turn-helix motifs are parallel to one another and fit within adjacent major grooves on DNA. The recognition helix (α6) in the first helix-turn-helix motif in the DNA-binding domain of UFA-bound ToxT is turned perpendicular to the recognition helix in the second helix-turn-helix motif (α9), which prevents it from fitting within the major groove of DNA (Fig. 1d). These results suggest UFAs inhibit both dimerization and DNA-binding of ToxT by controlling the positions of α3 and α6, respectively.

The similarity of the ToxT$_{EPI}$ and ToxT$_{ENV256}$ structures indicate that ToxT from the non-epidemic *V. cholerae* strain SCE-256 functions by the same mechanism as ToxT from epidemic *V. cholerae* serotype O1 El Tor. Purified ToxT$_{ENV256}$ is more soluble and crystallizes more readily than ToxT$_{EPI}$. For this reason, all further structural and biochemical experiments in this study were performed using ToxT$_{ENV256}$.

**ToxT$_{ENV256}$ K231A is resistant to unsaturated fatty acids**. Although mutation of Lys230 to alanine in ToxT$_{EPI}$ was shown to increase the ToxT-dependent expression of *ctx*[23], it has not been shown biochemically that the UFA-binding pocket seen in the crystal structure is responsible for regulating the activity of ToxT. To confirm that the UFA-binding pocket is involved in regulating the activity of ToxT, ToxT$_{ENV256}$ with an alanine substituted for the lysine in the DNA-binding domain that contacts the carboxylate of the bound UFA (K231A) was purified. Purified ToxT$_{ENV256}$ K231A has the same secondary structure as wild-type ToxT$_{ENV256}$ and shows similar binding to the *tcpA* promoter in vitro (Supplementary Fig. 1). However, ToxT$_{ENV256}$ K231A was less sensitive to oleic acid, binding to DNA in the presence of a concentration of oleic acid that completely prevented the wild-type protein from binding (Fig. 2). ToxT$_{ENV256}$ K231A also has lower thermostability than wild-type ToxT$_{ENV256}$, which is consistent with the hypothesis that the mutant would have a lower affinity for UFA (Fig. 2c). These results support the hypothesis that the UFA-binding pocket containing Tyr13, Lys32, and Lys231 of ToxT$_{ENV256}$, and Tyr12, Lys31, and Lys230 of ToxT$_{EPI}$, is responsible for the allosteric regulation of ToxT activity by UFAs.

**Structure reveals a mechanism for regulation of dimerization**. Although it has not been possible to produce diffracting crystals of ToxT$_{EPI}$ UFA-binding pocket mutants, leveraging the increased solubility of ToxT$_{ENV256}$ allowed us to obtained diffraction quality crystals of ToxT$_{ENV256}$ K231A. Crystals of

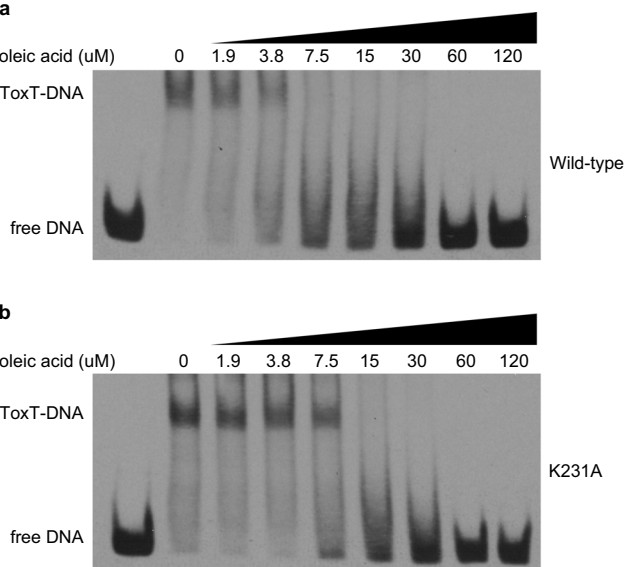

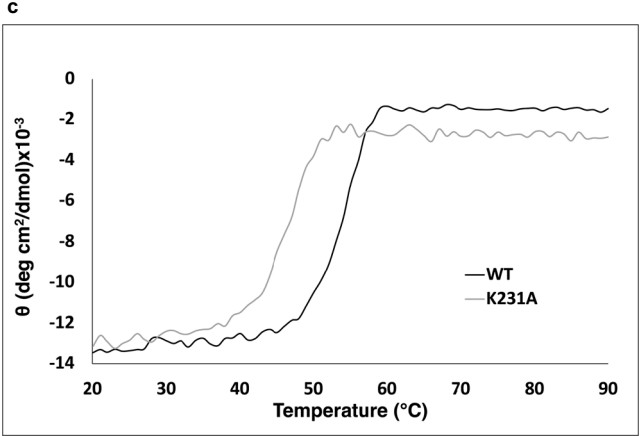

**Fig. 2** ToxT$_{ENV256}$ K231A is less sensitive to oleic acid and has reduced thermostability. Electrophoretic mobility shift assays showing the effect of oleic acid on DNA binding of ToxT$_{ENV256}$ wild-type (**a**) and K231A (**b**). All lanes contain 9 nM DIG-labeled 84 bp segment of dsDNA containing the ToxT-binding sites from the *tcpA* promoter DNA and 0.78 μM ToxT$_{ENV256}$. The concentration of oleic acid in each lane is shown. **c** Thermostability of wild-type (black) and K231A (gray) ToxT$_{ENV256}$ as determined by circular dichroism.

ToxT$_{ENV256}$ K231A were produced in the same crystallization condition as wild-type ToxT$_{ENV256}$ and belonged to the same space group with a single monomer in each asymmetric unit. However, a subset of crystals was found to have slightly different unit cell dimensions (Table 1). The first subset of ToxT$_{ENV256}$ K231A crystals, with the same unit cell dimensions as wild-type ToxT$_{ENV256}$, contained a UFA within the binding pocket, as obvious positive density in the shape of a UFA can be seen (Supplementary Fig. 2). Remarkably, no such density is seen in the structure of ToxT$_{ENV256}$ K231A solved using diffraction data from the second subset of crystals, leading to the conclusion that this subset of crystals contained apo ToxT$_{ENV256}$.

Comparison of the apo ToxT$_{ENV256}$ structure with the structure of UFA-bound ToxT$_{ENV256}$ revealed conformational changes that occur upon the release of UFA (Fig. 3). While no conformational differences in the DNA-binding domain are seen between the UFA-bound and apo structures, differences are seen in the regulatory domain. Amino acids Thr138, Gln139, Tyr140,

**Table 1 Crystallographic data collection and refinement statistics.**

| | UFA-bound WT ToxT$_{ENV256}$ 6P7R | UFA-bound ToxT$_{ENV256}$ K231A 6PB9 | Apo ToxT$_{ENV256}$ K231A 6P7T |
|---|---|---|---|
| Data collection | | | |
| Space group | C 1 2 1 | C 1 2 1 | C 1 2 1 |
| Cell dimension | | | |
| a, b, c (Å) | 79.35, 47.13, 74.56 | 79.3, 46.9, 74.4 | 79.8, 45.5, 77.0 |
| α, β, γ (°) | 90, 98.17, 90 | 90, 98.2, 90 | 90, 97.70, 90 |
| Resolution (Å) | 28.3–1.8 (1.86–1.80) | 28.21–2.11 (2.18–2.11) | 26.55–2.50 (2.59–2.50) |
| $R_{merge}$ | 0.054 (0.517) | 0.072 (0.687) | 0.0828 (0.697) |
| $R_{pim}$ | 0.030 (0.295) | 0.030 (0.282) | 0.0347 (0.292) |
| $I/\sigma(I)$ | 13.73 (2.07) | 14.54 (2.36) | 12.35 (2.81) |
| $CC_{1/2}$ | 0.998 (0.780) | 0.999 (0.818) | 0.997 (0.859) |
| Completeness (%) | 99.4 (99.92) | 97.97 (92.81) | 99.82 (99.69) |
| Redundancy | 4.1 (4.0) | 6.7 (6.6) | 6.7 (6.6) |
| Refinement | | | |
| Resolution (Å) | 28.3–1.8 (1.9–1.8) | 28.2–2.1 (2.2–2.1) | 26.6–2.5 (2.6–2.5) |
| Unique reflections | 25330 (2508) | 15477 (1458) | 9665 (960) |
| $R_{work}/R_{free}$ | 0.178/ 0.222 | 0.23/0.28 | 0.246/0.287 |
| # protein atoms | 2263 | 2250 | 2217 |
| # ligands atoms | 24 | 18 | 0 |
| Average B-factor | 46.11 | 53.54 | 80.83 |
| RMS bonds (Å) | 0.008 | 0.006 | 0.005 |
| RMS angles (°) | 1.16 | 1.11 | 0.98 |

and Ser141, that form part of the loop between helix α2 and helix α3 in the UFA-bound structure, form an additional turn of helix α3 in the apo structure (Fig. 3b). In addition, residues Leu108, Tyr109, Asn110, Glu111, and Lys112 form a new helix under helix α3 (Fig. 3c), and a new salt-bridge between Arg96 and Glu157 is formed that may stabilize the position of helix α3 in the apo state (Fig. 3d). In apo ToxT$_{ENV256}$, helix α3 is more parallel with helix α2 and is in better alignment with the structure of the dimerization domain of AraC (Fig. 3e). Taken together, the structures of UFA-bound and apo ToxT$_{ENV256}$ suggest that UFAs allosterically regulate the dimerization of ToxT by altering the position and length of helix α3.

While it is known that homodimerization is required for ToxT to bind DNA and activate the expression of virulence genes[10,11], the crystals of apo ToxT$_{ENV256}$ contain only a monomer in each asymmetric unit and there are no lattice contacts that suggest the location of the dimer interface. It is possible that full-length ToxT is only able to dimerize once bound to DNA. Using the structures of the AraC regulatory domain dimer and the MarA–DNA complex, a model of the ToxT homodimer bound to DNA was generated (Fig. 4a). This model suggests that when ToxT$_{ENV256}$ dimerizes, Lys158 and Asp143 on helix α3 of each subunit form salt bridges with Asp143 and Lys158, respectively, on helix α3 of the opposing subunit. In addition, the sidechain of Ile157, located near the middle of helix α3, would pack against Gly151 of the opposing subunit. Asp143, Ile147, Gly151, and Lys158 of ToxT$_{ENV256}$ are analogous to Glu142, V146, Gly150, and Lys157 of ToxT$_{EPI}$. To establish that helix α3 constitutes the interface of the ToxT dimer, LexA-fusion assays were performed using the dimer domains of ToxT$_{EPI}$ and ToxT$_{ENV256}$. LexA is capable of repressing the expression of sulA only when it is able to dimerize and fusion of putative dimerization domains to the DNA-binding domain of LexA has been used previously to demonstrate the dimerization of ToxT$_{EPI}$[10,11,32,33]. The introduction of a charge repulsion by mutating Lys157 to a glutamate, or mutating Gly150 to a leucine to sterically clash with the valine on the opposing subunit, disrupts dimerization of the regulatory domain of ToxT$_{EPI}$ in vivo (Fig. 4b). Similar results were obtained for ToxT$_{ENV256}$ (Supplementary Fig. 3a). Not surprisingly, mutation of Lys230 in ToxT$_{EPI}$ or Lys231 in ToxT$_{ENV256}$ to

alanine was not sufficient to promote dimerization of full-length ToxT$_{ENV256}$ in the bacterial two-hybrid assay. These results confirm that helix α3 is a crucial component the ToxT homodimer interface.

To confirm that dimerization at the helix α3 interface is necessary for DNA-binding, electrophoretic mobility shift assays were performed with ToxT$_{ENV256}$ K158E. Purified ToxT$_{ENV256}$ K158E has an identical secondary structure to wild-type ToxT$_{ENV256}$ (Supplementary Fig. 3b). However, ToxT$_{ENV256}$ K158E is unable to bind to DNA in vitro (Fig. 4c). While the addition of wild-type ToxT$_{ENV256}$ results in a decrease in the electrophoretic mobility of double-stranded DNA (dsDNA) containing the ToxT-binding site of the tcpA promoter, and a specific band of ToxT–DNA complex is seen, no specific ToxT-DNA band appears with the addition of ToxT$_{ENV256}$ K158E, indicating only nonspecific binding. These results indicate that the helix α3 dimer interface is required for DNA binding by ToxT.

**Allosteric control of dimerization via altered flexibility.** The absence of a pathway of conformational change between the dimerization helix of ToxT$_{ENV256}$ and the UFA-binding pocket suggests that UFA-binding controls dimerization by a dynamics-based allosteric mechanism. As the UFA-bound ToxT$_{ENV256}$ and apo ToxT$_{ENV256}$ structures were solved from the same crystal condition and space group, with the same lattice contacts, crystallographic B-factors were used to analyze the dynamics of the protein before and after the release the UFA ligand. The mean B-factor of the α-carbons in the apo-ToxT$_{ENV256}$ structure is 40 Å$^2$ higher than that of the UFA-bound structure. Furthermore, the regions of apo ToxT$_{ENV256}$ with the largest increase in normalized B-factors are loop 1 between β1 and β2, loop 7 between β7 and β8, helix α2, and helix α6 (Fig. 5 and Supplementary Fig. 5). Loops 1 and 7 are directly adjacent to the UFA-binding pocket. Helix α2 links the β-barrel to the dimer helix α3. Helix α6 is the recognition helix in the first helix-turn-helix motif of the DNA-binding domain that is not in a position that would allow it to fit within the major groove of DNA. Conversely, the normalized B-factors for the residues that form the additional turn on helix

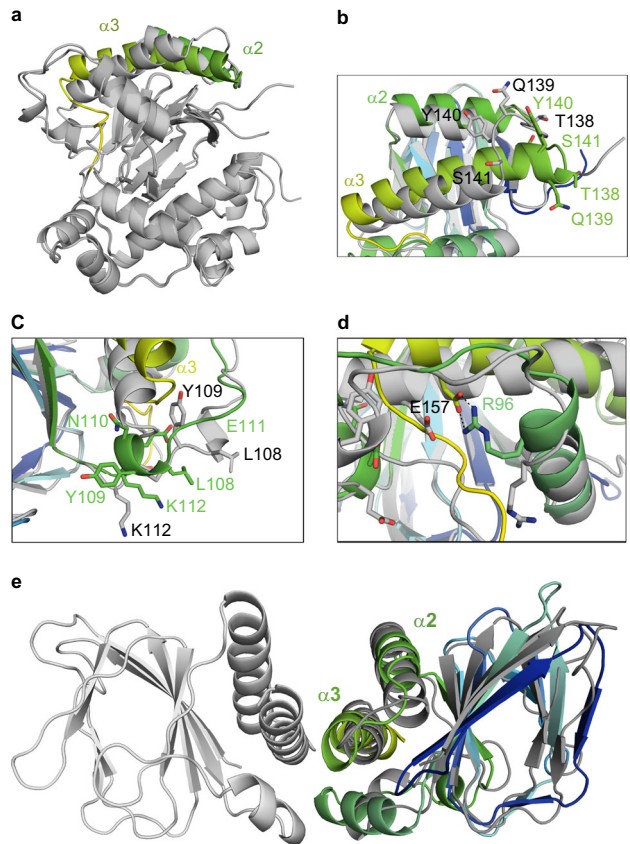

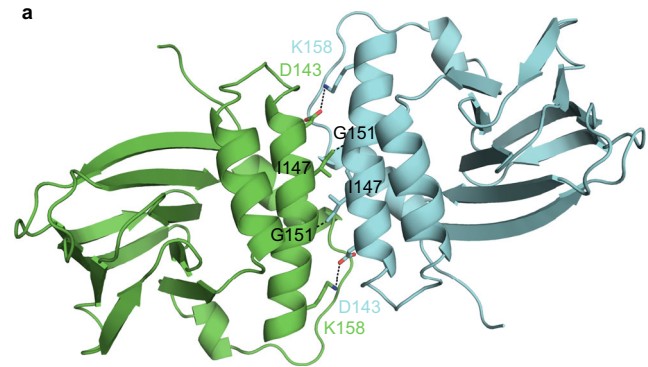

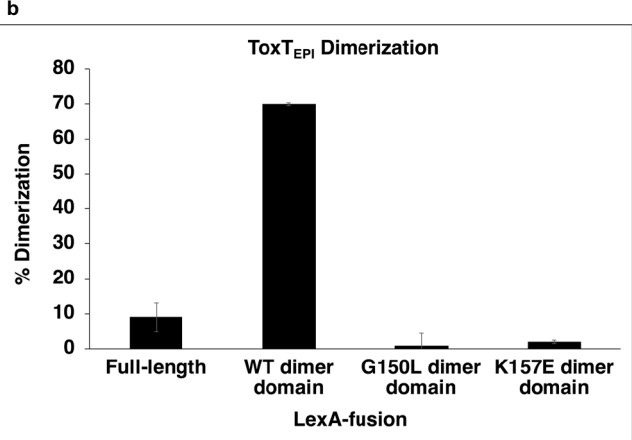

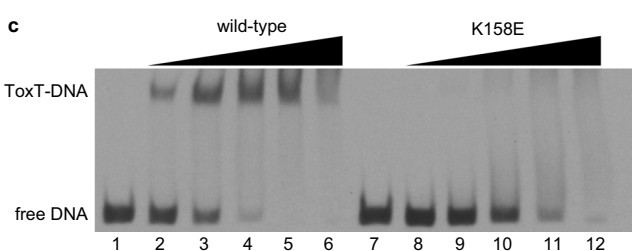

**Fig. 3** Structure of apo ToxT$_{ENV256}$. **a** Alignment of the crystal structure of apo ToxT$_{ENV256}$ (6P7T), dark gray, with the structure of UFA-bound ToxT$_{ENV256}$ (6PB9), light gray. Helices α2 and α3 of apo ToxT$_{ENV256}$ are colored green. **b** Close-up view of helices α2 and α3 of ToxT$_{ENV256}$. UFA-bound ToxT$_{ENV256}$ is colored light gray. Apo ToxT$_{ENV256}$ is colored blue to green. **c** Close-up view of the new helix in apo ToxT$_{ENV256}$ located between helix α1 and β9. UFA-bound ToxT$_{ENV256}$ is colored light gray. Apo ToxT$_{ENV256}$ is colored blue to green. **d** Close-up view of the new salt-bridge in apo ToxT$_{ENV256}$ between helices α1 and α3. UFA-bound ToxT$_{ENV256}$ is colored light gray, apo ToxT$_{ENV256}$ is colored blue to green. Amino acids of UFA-bound ToxT$_{ENV256}$ are labeled in black, amino acids of apo ToxT$_{ENV256}$ are labeled in green. **e** Structural alignment of the regulatory domain of apo ToxT$_{ENV256}$ to the AraC regulatory domain dimer. AraC is colored gray, the regulatory domain of apo ToxT$_{ENV256}$ is colored blue to green.

α3 and the new helix under helix α3 decrease after the release of UFA. The analysis of the crystallographic B-factors before and after UFA release indicates that ToxT bound to UFA is held in a tense state that is unable to dimerize or bind DNA. The release of UFA causes an increase in the flexibility of the protein that allows ToxT to adopt a relaxed conformation in which dimerization and DNA binding are possible.

**SAXS model of the ToxT dimer–DNA complex.** We have shown that in order to bind DNA, ToxT must form a symmetric homodimer through contacts between helix α3 of each subunit. However, in the model of a closed ToxT dimer, the DNA-binding helices of each subunit are positioned on the outside of the dimer, at a distance from one another greater than the distance between pairs of ToxT-binding sites on the *ctx* or *tcpA* promoters. To bind adjacent sites on DNA, a ToxT dimer would have to adopt an open conformation, with the DNA-binding domains separating from the dimerization domains. Mutations predicted to force ToxT into an "open" conformation have been shown to enhance

**Fig. 4** Dimerization of ToxT at the helix α3 interface is required for DNA binding. **a** Model of the ToxT dimer interface. **b** LexA-fusion bacterial two-hybrid dimerization assay of ToxT$_{EPI}$ dimer interface mutants. Error bars indicate standard deviation. A western blot of the LexA-ToxT$_{EPI}$ fusions confirming expression is shown in Supplementary Fig. 4. **c** Electrophoretic mobility shift assay of wild-type and K158E ToxT$_{ENV256}$ binding to the *tcpA* promoter. All lanes contain 9 nM DIG-labeled 84 bp segment of dsDNA containing the ToxT-binding sites from the *tcpA* promoter. Lane 1, free DNA; lane 2, 0.098 μg wild-type ToxT$_{ENV256}$; lane 3, 0.195 μM wild-type ToxT$_{ENV256}$; lane 4, 0.39 μM wild-type ToxT$_{ENV256}$; lane 5, 0.78 μM wild-type ToxT$_{ENV256}$; lane 6, 1.56 μM wild-type ToxT$_{ENV256}$; 7, free DNA; lane 8, 0.098 μM K158E ToxT$_{ENV256}$; lane 9, 0.195 μM K158E ToxT$_{ENV256}$; lane 10, 0.39 μM K158E ToxT$_{ENV256}$; lane 11, 0.78 μM K158E ToxT$_{ENV256}$; lane 12, 1.56 μM K158E ToxT$_{ENV256}$. Error bars are indicated (n = 3 experiments). A western blot of the LexAToxTEPI fusions confirming expression of protein has been provided as Supplementary Fig. 4.

ToxT activity in the presence of UFAs and inhibitors[11]. A homology model of an open ToxT dimer bound to DNA was constructed by aligning the N-terminal domains of each ToxT subunit to the structure of the AraC regulatory domain dimer, and the DNA-binding domains to two copies of the MarA-DNA structure (Fig. 6a). The regulatory and DNA-binding domains of each subunit of the dimer remain connected by the linker. In the

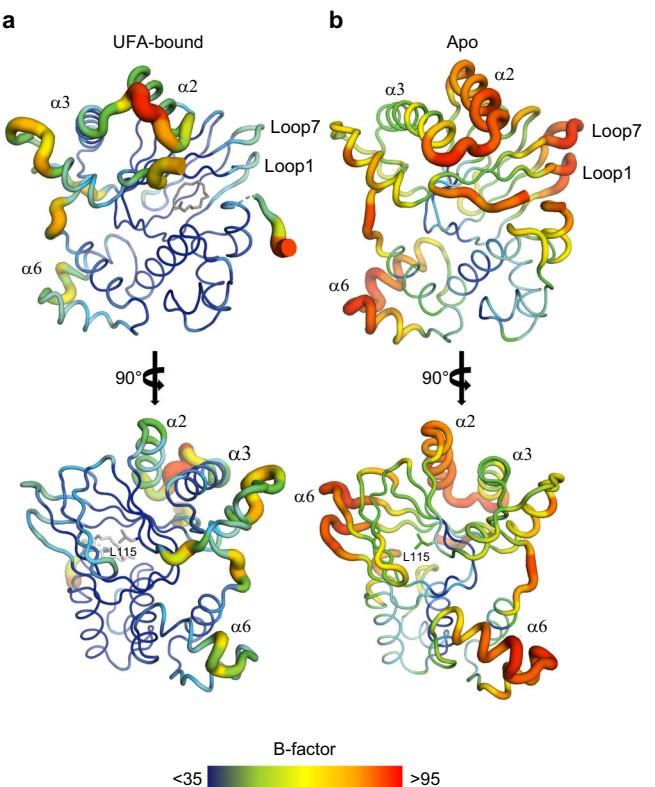

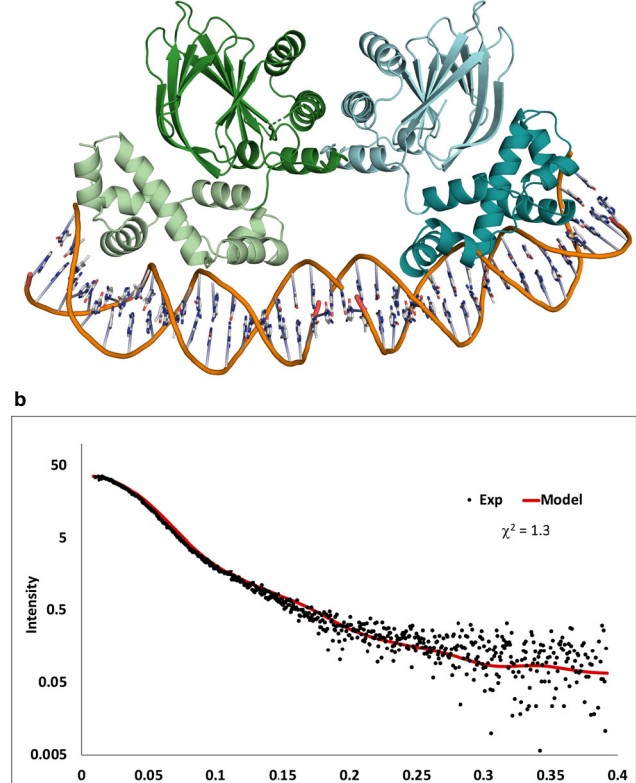

**Fig. 5** Dynamics of ToxT$_{ENV256}$ increase in the absence of bound UFA. Crystallographic B-factors are indicated in putty thickness and color from dark blue to red. **a** B-factor putty representation of UFA-bound ToxT$_{ENV256}$. **b** B-factor putty representation of apo ToxT$_{ENV256}$.

**Fig. 6** SAXS validation of a model of the open ToxT dimer–DNA complex. **a** Model of an open ToxT dimer bound to bent DNA. **b** Inline SEC-SAXS data recorded on the purified ToxT–*ctx* complex. Experimental data are shown in black. Scattering data calculated from the model in **a** are shown in red.

model, each DNA-binding domain would bend the DNA by 35° as seen with MarA-DNA. To confirm this model, the ToxT–*ctx* complex was purified (Supplementary Fig. 6) and inline size exclusion chromatography-small-angle X-ray scattering (SEC-SAXS) data was collected (Fig. 6b). The calculated scattering plot of the ToxT-DNA model fits the experimental scattering data with a $\chi^2$-value of 1.3. Calculated scattering plots for models of a single open-monomer bound to DNA, two closed monomers bound to DNA, or an open dimer bound to straight DNA do not fit the experimental data as well (Supplementary Fig. 7). Therefore, our SAXS results are consistent with a model of a symmetric open dimer bound to bent DNA.

## Discussion

We have presented here the results of structural and functional studies of ToxT that reveal the detailed molecular basis for the regulation of virulence in *V. cholerae* by the UFA components of bile or synthetic ToxT inhibitors. The increased solubility of ToxT from *V. cholerae* serogroup O42 strain SCE-256 (ToxT$_{ENV256}$) facilitated the crystallization of ToxT free of inhibitor. Crystals of UFA-bound and apo ToxT$_{ENV256}$ were obtained in the same solution conditions and space group, allowing direct comparison of the dynamics of the ligand-bound and apo conformational states using crystal data. Small-angle X-ray scattering then facilitated the accurate modeling of an active ToxT dimer bound to the cholera toxin promoter.

The crystal structures presented here of UFA-bound and apo ToxT$_{ENV256}$ reveal that UFAs and synthetic inhibitors regulate ToxT dimerization by controlling the position and length of helix α3. We have provided evidence that helix α3 forms the interface of the ToxT homodimer and that dimerization at this interface is

necessary for ToxT to bind DNA. However, no obvious pathway of structural change linking the UFA-binding pocket to the dimerization helix is observed. Therefore, we propose that an increase in protein flexibility occurs when ToxT is free of UFA or inhibitor, and that this increase in flexibility allows ToxT to adopt a conformation in which dimerization and DNA-binding is possible.

These results suggest a model of dynamic allosteric regulation of ToxT dimerization and DNA-binding by UFAs and inhibitors (Fig. 7). In this model, ToxT bound to a UFA or inhibitor is trapped in a conformation that is unable to dimerize or bind DNA. Upon the release of the UFA or inhibitor, ToxT is more relaxed and is allowed to sample a conformation in which dimerization on DNA is possible. A dynamic model for the allosteric regulation of ToxT by UFAs and inhibitors could explain why the mutation of a leucine at position 114 in ToxT$_{EPI}$ to either a proline or alanine confers resistance to UFAs and virstatin and allows the full-length protein to dimerize[16,17]. Leucine 114 of ToxT$_{EPI}$ is located with its hydrophobic sidechain pointed into the back of the UFA-binding pocket at the beginning of the β-sheet preceding helices α2 and α3. The sidechain of Leu61, which has also been shown to increase ToxT activity when mutated to an alanine[23], is between the sidechain of Leu114 and the UFA (Supplementary Fig. 8). When bound to ToxT, a UFA or inhibitor makes hydrophobic contact with Leu61, which makes contact with Leu114. In this state, ToxT is held in a conformation that is not able to dimerize. When the UFA or inhibitor is released from ToxT, the interaction between the UFA and Leu114, via Leu61, would be lost. The loss of this interaction

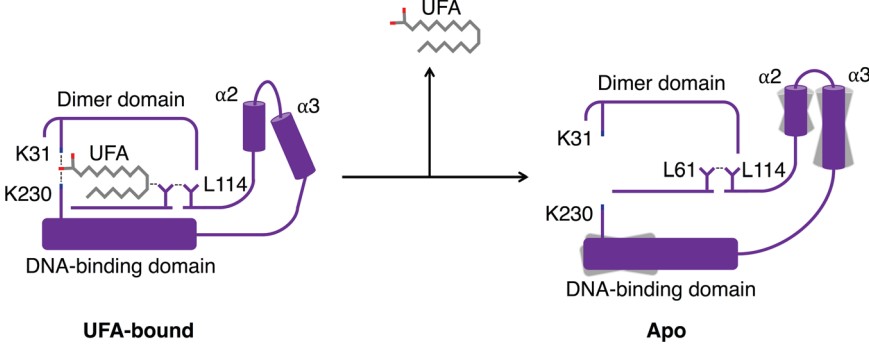

**Fig. 7** Proposed mechanism for the allosteric regulation of ToxT$_{EPI}$ dimerization by UFAs. Left: In the UFA-bound state, hydrophobic contacts between the UFA, Leu61 and L114, trap ToxT in a tense state in which the conformation of helix α3 precludes dimerization. Right: Release of UFA from ToxT relaxes helices α2 and α3, which are then able to adopt a conformation allowing dimerization on DNA.

results in an increase in flexibility beginning at Leu114, continuing through helix α2 into the dimer helix, allowing helix α3 to move into a position that allows dimerization. This model may also explain why the loss of the DNA-binding domain promotes the dimerization of the ToxT regulatory domain in vivo (Fig. 4b)[10,16,17]. As the C-terminus of helix α3 is directly linked to the DNA-binding domain, truncation of the DNA-binding domain may also increase the flexibility of helix α3, allowing it to adopt a conformation that is able to dimerize.

While we see no conformational change in the DNA-binding domain upon the release of UFA, we cannot eliminate the possibility that UFAs also directly control ToxT DNA binding. The crystallographic B-factors of recognition helix α6 are elevated in the UFA-free structure and this increase in flexibility may allow the helix to change conformation upon association with DNA. Regardless, the results presented here demonstrate that the regulation of dimerization is sufficient to control DNA binding by ToxT.

Before the crystal structure of ToxT$_{EPI}$ was determined, it was proposed that ToxT binds to sites on DNA that can be arranged as either direct or inverted pairs of ToxT-binding sites[12]. The ToxT-binding sites in the *ctx* and *tcpA* promoters were proposed to be direct. However, our results indicate that, like AraC, the regulatory domain of ToxT forms dimers with twofold symmetry. Furthermore, the linker connecting the dimer helix to the DNA-binding domain threads through the loop between helix α1 and β9, limiting its reach. Our SAXS results strongly support a model of an open and symmetric ToxT dimer bound to both the *ctx* and *tcpA* promoters.

In conclusion, we propose the following detailed model for the regulation of virulence gene expression in *V. cholerae* by the UFAs found in bile (Fig. 7): When *V. cholerae* is in the lumen of the small intestine in the presence of high concentrations of bile, ToxT is bound to a UFA and is trapped in an inactive conformation that is unable to dimerize or bind DNA and virulence genes are not expressed. Once the bacterium penetrates the mucus layer and moves to the epithelial surface, where the concentration of bile is lower, UFA is released, resulting in an increase in flexibility that allows the regulatory domain to sample a conformation in which ToxT can dimerize. Additionally, increased flexibility in the DNA-binding domain allows both recognition helices to fit within adjacent major grooves of DNA. ToxT then binds to DNA as an open symmetric dimer and activates the expression of virulence genes.

## Methods
**Purification of wild-type and mutant ToxT$_{ENV256}$**. Wild-type or mutant ToxT from *Vibrio cholerae* strain SCE256 (ToxT$_{ENV256}$) was cloned into plasmid pTXB1 (New England Biolabs) to generate a ToxT$_{ENV256}$-intein/Chitin-binding domain

construct. The construct was expressed in BL21(DE3) codonplus-RIL cells (Agilent Technologies) induced by autoinduction in ZYM-5052 media overnight at 20 °C (Studier, 2005). All Luria-Bertani (LB) agar plates and media contained 100 μg/ml carbenicillin and 25 μg/ml chloramphenicol. Cells were lysed by sonication in lysis buffer (20 mM Tris HCl pH 8, 500 mM NaCl, 1 mM EDTA) at 4 °C and centrifuged at 120,000 × g for 30 min. The supernatant was filtered using a 0.45 μm filter and loaded by gravity onto a column packed with chitin resin (New England Biolabs). The column was washed with column buffer (20 mM Tris HCl pH 8, 200 mM NaCl, 1 mM EDTA) then washed with cleavage buffer (20 mM Tris HCl pH 8, 200 mM NaCl, 1 mM EDTA, 100 mM DTT) before being incubated overnight at 4 °C. ToxT$_{ENV256}$ was eluted from the chitin resin using column buffer. Eluent from the chitin column was loaded onto a Hitrap SP cation exchange column (GE Life Sciences) using an AKTA Explorer FPLC system. ToxT$_{ENV256}$ was eluted from the Hitrap SP column with a linear gradient of column buffer with 200 mM-1 M NaCl.

**Crystallization of wild-type and K231A ToxT$_{ENV256}$**. Purified wild-type ToxT$_{ENV256}$ or ToxT$_{ENV256}$ K231A was concentrated to 5 mg/ml using Amicon Ultra centrifugal filter units. Crystal conditions were screened by sitting drop vapor diffusion. Single-diffraction quality crystals of wild-type ToxT$_{ENV256}$ and ToxT$_{ENV256}$ K231A were obtained by mixing equal volume of protein and 0.2 M sodium citrate tribasic dihydrate, 20% w/v polyethylene glycol 3350. Crystals appeared within an hour and grew to their maximum size overnight. The crystallization solutions supplemented with 40% glycerol were used as cryo-protectants, and crystals were flash frozen in liquid nitrogen.

**Data collection and processing**. X-ray diffraction data was collected at the FMX beamline National Synchrotron Light Source II (NSLSII), Brookhaven National Laboratory, Upton, NY. A 1.8 Å data set of 1800 frames with an oscillation range of 0.2° was collected at a wavelength of 0.9790 Å with 0.1 s exposures at 100° K. The crystal to detector distance was 220 mm. The data set was indexed, integrated, scaled and merged using *XDS*[34]. Data collection statistics are shown in Table 1.

**Structure determination and refinement**. The reflection file was converted and $R_{free}$ flags set (10% of unique reflections) using Phenix reflection file editor[35]. The Matthew's coefficient was calculated, and it was determined that the asymmetric unit contained a single dimer of ToxT$_{ENV256}$. The structure of ToxT$_{ENV256}$ was solved by molecular replacement using Phenix Phaser-MR with ToxT$_{EPI}$ (3GBG) as the search model[36]. Multiple rounds of refinement were carried out using Coot and Phenix.refine[37,38]. Refinement statistics are shown in Table 1. Structural figures were generated using PyMOL[39].

**Circular dichroism**. Purified ToxT$_{ENV256}$ was dialyzed into CD buffer (10 mM Tris pH 8, 150 mM NaCl) and diluted to 10–15 μM. CD scans were acquired at 20 °C with three accumulations each in the 190–250 nm range at 100 nm/min with a 1 nm bandwidth. CD melting curves were collected at 222 nm between 20 °C and 90 °C with a ramp rate of 1 °C/min.

**Electrophoretic mobility shift assay**. A 5'-digoxigenin (DIG) labeled 84-bp dsDNA fragment containing the ToxT-binding sequence of the *tcpA* promoter from *V. cholerae* strain O395 was generated by PCR using 5'-DIG-labeled primers as previously described[24]. Purified WT or mutant ToxT$_{ENV256}$ was mixed with DIG-labeled dsDNA in EMSA buffer (10 mM Tris pH 7.5, 1 mM EDTA, 100 mM KCl, 5 mM MgCl$_2$, 1 mM DTT, 0.3 mg/ml BSA, 0.25 mg/ml poly[d(I-C)], 10% glycerol) and incubated at 30 °C for 15 min. Reactions were loaded on a 5% polyacrylamide TBE gel and subject to electrophoresis in chilled 0.75× TBE. The labeled DNA was transferred onto a positively charged nylon membrane by

electroblotting in 0.5× TBE at 4 °C. After UV cross-linking, DIG-labeled DNA was probed with an Alkaline phosphatase conjugated anti-DIG antibody, developed with CSPD and exposed to X-ray film.

**LexA-fusion bacterial 2-hybrid assay.** ToxT$_{EPI}$ and ToxT$_{ENV256}$ constructs were cloned in plasmid pSR662 and transformed into *sulA-lacZ E. coli* strain SU101. Overnight cultures of each strain were diluted into fresh LB pH 6.5 containing 1 mM IPTG and grown for 4 h at 30 °C. β-galactosidase activity was quantified as described[32,33]. Western blots were performed using an anti-LexA-DNA-binding domain antibody.

**Small-angle X-ray scattering.** The ToxT$_{ENV256}$–DNA complex was reconstituted by mixing purified ToxT$_{ENV256}$ and a 33-bp dsDNA fragments containing the ToxT-binding sites from the *ctx* promoters at 3:1 protein: DNA and dialyzing into binding buffer (10 mM HEPES pH 7, 100 mM NaCl, 1 mM EDTA, 40 mM sodium bicarbonate) at room temperature. The complexes were then purified by size exclusion chromatography using a Superdex 200 10/300 and an AKTA Explorer FPLC system in binding buffer. Inline size exclusion chromatography small-angle X-ray scattering (SEC-SAXS) data of the purified ToxT$_{ENV256}$–DNA complexes were collected at the SIBYLS beamline at the Advanced Light Source at Lawrence Berkeley National Laboratory[40,41]. Buffer subtraction and merging of scattering data was performed using SCÅTTER[42]. The calculated SAXS profiles for the ToxT-DNA models were generated and fit to the experimental data using FoXS[43].

**Reporting summary.** Further information on research design is available in the Nature Research Reporting Summary linked to this article.

## Data availability

Coordinates and structure factors have been submitted to the Protein Data Bank under accession numbers: 6P7R, 6P7T, and 6PB9.

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

## Acknowledgements

This study was supported by National Institutes of Health (NIH) awards AI072661 (F.J.K.), AI025096 (K.S.), and AI120068 (K.S. and F.J.K.). This work was supported in part by grant P20GM113132 from NIGMS. X-ray data were collected at: the Life Science Biomedical Technology Research resource, which is primarily supported by the National Institute of Health, National Institute of General Medical Sciences (NIGMS) through a Biomedical Technology Research Resource P41 grant (P41GM111244), and by the DOE Office of Biological and Environmental Research (KP1605010); and at the Advanced Light Source (ALS), a national user facility operated by Lawrence Berkeley National Laboratory on behalf of the Department of Energy, Office of Basic Energy Sciences, through the Integrated Diffraction Analysis Technologies (IDAT) program, supported by DOE Office of Biological and Environmental Research (additional support comes from the National Institute of Health project ALS-ENABLE (P30 GM124169) and a High-End Instrumentation Grant S10OD018483).

## Author contributions

J.T.C., K.S., and F.J.K. conceived of the study; J.T.C. performed all protein purifications, crystallization, structure determination, structural analysis, EMSA's, SAXS experiments and data analysis, generated the figures and wrote the manuscript; K.S. and F.J.K. edited the manuscript; G.K. designed E.M.S.A. and LexA experiments; K.A.C. performed mutagenesis and LexA assays; A.K.W. cloned wild-type ToxTENV256 for purification.

## Competing interests

The authors declare no competing interests.
