## [Peer Review File · Communications Biology]

Reviewers' comments:

Reviewer #1 (Remarks to the Author):

The manuscript by Cruite et al describes the structure of the virulence regulator ToxT of *Vibrio cholerae* in the absence of fatty acid, in order to understand the mechanism of fatty acid regulation of ToxT activity. The trick was to use ToxT from an environmental *V. cholerae* strain, which was more soluble, and then introduce a mutation within the fatty acid binding pocket to inhibit binding. They solved the structure of this protein +/- fatty acid, which allowed for determining the effect of fatty acid binding on ToxT structure to be resolved. Interestingly the absence of fatty acid led to changes in the N-terminal regulatory domain that repositioned one of the alpha helices ($\alpha 3$), and the authors present evidence that this would lead to the flexibility to allow the protein to dimerize and bind DNA. Modeling and X-ray scattering studies were consistent with the apo form of the protein forming symmetric dimers binding the promoter and bending the DNA. These are extremely important results that document the molecular mechanism of bile-mediated modulation of ToxT transcriptional activation, and hence *V. cholerae* virulence. The manuscript was well-written and clear, with experimentation provided to back up key elements of the hypotheses that followed from the crystal structures. ToxT-dependent transcription is at the heart of cholera pathogenesis, so these results are of immense importance in understanding not only *V. cholerae* virulence, but also transcription activation by AraC-like regulatory proteins.

Reviewer #2 (Remarks to the Author):

This manuscript reports a crystal structure of the *Vibrio cholerae* ToxT master virulence activator, belonging to the AraC/XylS family of transcriptional regulator. The structure suggests an inhibition mechanism where a release of unsaturated fatty acid (UFA) or inhibitor from the regulator increases the flexibility and allows ToxT to dimerize and bind target DNA. The authors further use small angle X-ray scattering to validate a structural model of the ToxT dimer bound to the cholera toxin promoter. This is a nice piece of work and should be acceptable for publication in *Commun. Biol.* There are a few very minor comments that the authors may want to consider before publication.

Minor comments:

p.3 lines 1-5: Why is that full-length ToxT cannot be dimerized, but dimerization is necessary for DNA binding? Can the authors use a few sentences to elaborate that?

p.5 2nd last line: There is a statement "... ToxT is turned nearly 90° relative to ...". Can the authors use an arrow in Fig. 1D to show how the helix turns 90°?

Fig. 1: Please label the secondary structural elements in Fig. 1A, C and D. The orientation of 1A, C, and D are quite different. It would be easier to understand the structural information by making these figures more or less in the same orientation.

p.15 lines 14-15: The statement "The mean B-factor ... upon release of UFA." There is no evidence that the release of UFA causes the increase in dynamics of the protein. The authors may want to rephrase this statement. How about "The mean B-factor of apo-ToxT is 40 Å² higher than that of the UFA-bound structure."

Reviewer #3 (Remarks to the Author):

ToxT is the master transcription factor that controls expression of two major virulence factors, cholera toxin and the toxin coregulated pilus in *Vibrio cholera*. ToxT is an AraC/XylS-family transcription factor, negatively regulated by unsaturated fatty acids (UFAs) in bile. The structure of the epidemic strain O1 El Tor ToxT in complex with palmitoleic acid and inhibitors have been solved previously, which all present an inactive form of ToxT, while the active-form structure of ToxT is unknown. In this manuscript, the authors investigated ToxT from another strain (ToxT-env256) and reported three structures: WT ToxT-env256 in complex with UFA, ToxT-env256 K231A in complex with UFA and ToxT-env256 K231A apo. By comparing the UFA-bound and apo ToxT-env256 K231A, they identified structural features of ToxT in the active conformation. Though no experimental ToxT-DNA-complex structure was determined, the authors provided a reasonable theoretical model of this complex, which was validated by SAXS experiment. Finally, the authors proposed a conformation-dynamics based activation mechanism of ToxT. Overall, the manuscript is well written and the results provide valuable information to understand the activation of ToxT. The following are specific questions/comments for the authors.

- (1)The ligand in the reported structure is modeled as an unsaturated fatty acid (UFA). What's the chemical nature of the UFA? The ligand should not be defined by simply referring to the ligand in the known ToxT structure. Experimental evidence is needed.
- (2)The orientation of UFA in ToxT-env256 is different from that in ToxT-EPI (3GBG) (Fig. 1a). Provided the two proteins share good similarity in structure, please explain and discuss in the maintext.
- (3)The orientations of UFA in K231A ToxT-env256 (Fig. S2a) and WT ToxT-env256 (Fig. 1b) are different. Please explain. An omit map for UFA in K231A ToxT-env256 should be provided. In the apo form, does the electron density in the cavity (Fig. S2b) represent water molecules? If so, please put them in the figure and indicate the hydrogen bonding.
- (4)The usage of ToxT-env256 is confusing, as it refers to both WT and mutants (eg, in Fig. S2) please specify WT or mutants in the manuscript, including the method section.
- (5)ToxT Dimerization is an important issue in this manuscript. Size exclusion chromatography of WT ToxT-env256, mutants, and their complexes with DNA should be provided.
- (6)The purified WT and K231A ToxT-env256 seemed to contain UFA, which were taken from the expression system. Nevertheless, these proteins were used in DNA binding experiment and behaved somehow normally. Please explain.
- (7)Fig. S4 is too lab-style. Please re-prepare in a formal format.
- (8)A couple of writing errors are found. A carefully check of the whole manuscript is needed. For examples, in Fig. S6, "Right, model, Left, calculated SAXS plot" should be "Left, model, Right, calculated SAXS plot"; in Fig. S1, "DIG-labeled DNA" should be "DIG-labeled DNA". The amount of protein is confusing in unit, ug or uM.

We thank the reviewers for their helpful remarks and thorough review of the manuscript. Below we address each of the comments/concerns raised.

Reviewers' comments:

Reviewer #1 (Remarks to the Author):

The manuscript by Cruite et al describes the structure of the virulence regulator ToxT of Vibrio cholerae in the absence of fatty acid, in order to understand the mechanism of fatty acid regulation of ToxT activity. The trick was to use ToxT from an environmental V cholerae strain, which was more soluble, and then introduce a mutation within the fatty acid binding pocket to inhibit binding. They solved the structure of this protein +/- fatty acid, which allowed for determining the effect of fatty acid binding on ToxT structure to be resolved. Interestingly the absence of fatty acid led to changes in the N-terminal regulatory domain that repositioned one of the alpha helices (a3), and the authors present evidence that this would lead to the flexibility to allow the protein to dimerize and bind DNA. Modeling and Xray scattering studies were consistent with the apo form of the protein forming symmetric dimers binding the promoter and bending the DNA.

These are extremely important results that document the molecular mechanism of bile-mediated modulation of ToxT transcriptional activation, and hence V. cholerae virulence. The manuscript was well-written and clear, with experimentation provided to back up key elements of the hypotheses that followed from the crystal structures. ToxT-dependent transcription is at the heart of cholera pathogenesis, so these results are of immense importance in understanding not only V. cholerae virulence, but also transcription activation by AraC-like regulatory proteins.

We thank Reviewer 1 for their positive remarks.

Reviewer #2 (Remarks to the Author):

This manuscript reports a crystal structure of the Vibrio cholerae ToxT master virulence activator, belonging to the AraC/XylS family of transcriptional regulator. The structure suggests an inhibition mechanism where a release of unsaturated fatty acid (UFA) or inhibitor from the regulator increases the flexibility and allows ToxT to dimerize and bind target DNA. The authors further use small angle X-ray scattering to validate a structural model of the ToxT dimer bound to the cholera toxin promoter. This is a nice piece of work and should be acceptable for publication in Commun. Biol. There are a few very minor comments that the authors may want to consider before publication.

We thank Reviewer 2 for their positive remarks and address each suggestion below.

Minor comments:

p.3 lines 1-5: Why is that full-length ToxT cannot be dimerized, but dimerization is necessary for DNA binding? Can the authors use a few sentences to elaborate that?

The following text has been added to the manuscript (new text in the manuscript is **in red**)
"While full-length ToxT has not been shown to form a dimer, the regulatory domain, when separated from the rest of the protein, has been shown to dimerize *in vivo*, suggesting that the interaction between the two domains somehow regulates dimerization."

p.5 2nd last line: There is a statement "... ToxT is turned nearly 90o relative to ...". Can the authors use an arrow in Fig. 1D to show how the helix turns 90o?

The following text has been added to the manuscript: "The recognition helix (α 6) in the first helix-turn-helix motif in the DNA-binding domain of UFA-bound ToxT is turned perpendicular to the recognition helix in the second helix-turn-helix motif (α 9), which prevents it from fitting within the major groove of DNA (Fig. 1d)."

Helix 9 was labeled in F1D.

Fig. 1: Please label the secondary structural elements in Fig. 1A, C and D. The orientation of 1A, C, and D are quite different. It would be easier to understand the structural information by making these figures more or less in the same orientation.

The relevant secondary structure elements in Fig. 1A, C and D were labeled.

p.15 lines 14-15: The statement "The mean B-factor ... upon release of UFA." There is no evidence that the release of UFA causes the increase in dynamics of the protein. The authors may want to rephrase this statement. How about "The mean B-factor of apo-ToxT is 40 Å² higher than that of the UFA-bound structure."

Text was modified as suggested: "The mean B-factor of the α -carbons in the apo-ToxT_{ENV256} structure is 40 Å² higher than that of the UFA-bound structure."

Reviewer #3 (Remarks to the Author):

ToxT is the master transcription factor that controls expression of two major virulence factors, cholera toxin and the toxin coregulated pilus in Vibrio cholera. ToxT is an AraC/XylS-family transcription factor, negatively regulated by unsaturated fatty acids (UFAs) in bile. The structure of the epidemic strain O1 El Tor ToxT in complex with palmitoleic acid and inhibitors have been solved previously, which all present an inactive form of ToxT, while the active-form structure of ToxT is unknown. In this manuscript, the authors investigated ToxT from another strain (ToxT-env256) and reported three structures: WT ToxT-env256 in complex with UFA, ToxT-env256 K231A in complex with UFA and ToxT-env256 K231A apo. By comparing the UFA-bound and apo ToxT-env256 K231A, they identified structural features of ToxT in the active conformation. Though no experimental ToxT-DNA-complex structure was determined, the authors provided a reasonable theoretical model of this complex, which was validated by SAXS experiment. Finally, the authors proposed a conformation-dynamics based activation mechanism of ToxT. Overall, the manuscript is well written and the results provide valuable information to understand the activation of ToxT. The following are specific questions/comments for the authors.

We thank the reviewer for their very thorough feedback on the manuscript. Below we describe how we have addressed each of the comments/concerns.

(1) The ligand in the reported structure is modeled as an unsaturated fatty acid (UFA). What's the chemical nature of the UFA? The ligand should not be defined by simply referring to the ligand in the known ToxT structure. Experimental evidence is needed.

While the copurified ligand was not extracted from ToxT_{ENV256} and characterized, we have done this previously for wild-type ToxT_{EPI}. Additionally, the UFA oleic acid was shown to inhibit DNA binding of ToxT_{ENV256}, similar to the result seen with ToxT_{EPI}. We feel that this is sufficient evidence that the copurified ligand is indeed a UFA, which was confirmed by NMR for the ToxT_{EPI} structure as described by our group and also observed in ToxT crystal structures determined by others (Li et al. Acta Crystallogr F Struct Biol Commun. 2016 Sep 1; 72(Pt 9): 726–731.)

(2) The orientation of UFA in ToxT-env256 is different from that in ToxT-EPI (3GBG) (Fig. 1a). Provided the two proteins share good similarity in structure, please explain and discuss in the main text.

This is an excellent observation and a subject of ongoing research. Structures we have determined that are not included in the manuscript indicate that leucine 25 in ToxT_{EPI} and valine 26 in ToxT_{ENV256} are responsible for determining the orientation of the bound UFA. However, because experiments have not yet been done to determine if the orientation of the UFA has any effect on the sensitivity of ToxT to UFA, this was not discussed in this manuscript. Note the small molecule inhibitors bound to ToxT as described in Woodbrey et al. Sci Rep. 2017; 7: 45011 also show flexible binding modes with respect to the hydrophobic region. Interestingly, the ToxT pocket appears to be somewhat plastic with respect to ligand binding. A separate manuscript addressing the variable orientation of the UFA is in preparation.

(3) The orientations of UFA in K231A ToxT-env256 (Fig. S2a) and WT ToxT-env256 (Fig. 1b) are different. Please explain. An omit map for UFA in K231A ToxT-env256 should be provided. In the apo form, does the electron density in the cavity (Fig. S2b) represent water molecules? If so, please put them in the figure and indicate the hydrogen bonding.

This was an error due to our investigating the phenomenon mentioned directly above and not updating the figure. The UFA is in the same orientation as the WT ENV.

While Fig. S2 shows $2f_o - f_c$ density, a simulated annealing composite omit map calculated using PHENIX shows nearly identical density in this area (contoured a 1.0 sigma).

Fig S2 has been corrected to include water molecules and hydrogen bonds.

(4) The usage of ToxT-env256 is confusing, as it refers to both WT and mutants (eg, in Fig. S2) please specify WT or mutants in the manuscript, including the method section.

“K231A” has been added to Fig S2 legend and throughout the manuscript to avoid this confusion.

(5) ToxT Dimerization is an important issue in this manuscript. Size exclusion chromatography of WT ToxT-env256, mutants, and their complexes with DNA should be provided.

A figure showing size exclusion chromatography of WT ToxT_{ENV256}, ctx DNA, and the ToxT_{ENV256}-ctx complex was added to the supplemental material (Fig. S6). The complex was confirmed to be a ToxT dimer bound to DNA by SAXS.

(6) The purified WT and K231A ToxT-env256 seemed to contain UFA, which were taken from the expression system. Nevertheless, these proteins were used in DNA binding experiment and behaved somehow normally. Please explain.

This has been somewhat puzzling to us for over a decade, but the EMSA results are clear and consistent, Furthermore, the small molecule ToxT-inhibitor structures solved in our lab show that UFA can be exchanged for other molecules (Woodbrey et al. Sci Rep. 2017; 7: 45011), therefore there must exist an equilibrium between ToxT ligand-bound and unbound states. We believe when UFA is present at low concentrations (i.e. equimolar as purified) ToxT binds to DNA (and likely releases UFA), but when additional UFA is added during an EMSA, the equilibrium shifts to ToxT-UFA and it releases from DNA.

(7) Fig. S4 is too lab-style. Please re-prepare in a formal format.

Fig. S4 has been replaced with a more finished version.

(8) A couple of writing errors are found. A carefully check of the whole manuscript is needed. For examples, in Fig. S6, “Right, model, Left, calculated SAXS plot” should be “Left, model, Right, calculated SAXS plot”; in Fig. S1, “DIG-labeled DNA” should be “DIG-labeled DNA”. The amount of protein is confusing in unit, ug or uM.

We have reviewed the manuscript and hopefully caught all such errors.

REVIEWERS' COMMENTS:

Reviewer #1 (Remarks to the Author):

The authors have addressed all comments sufficiently, and the revised manuscript is a nice piece of work.

Reviewer #2 (Remarks to the Author):

The authors have done a great job to address all minor issues from the previous submission. This manuscript should now be acceptable for publication in Communications Biology. Nice work!

Reviewer #3 (Remarks to the Author):

In the revised version, the authors have basically addressed the concerns and improved the manuscript.